# A Survey on 5G and LPWAN-IoT for Improved Smart Cities and Remote Area Applications: From the Aspect of Architecture and Security

**DOI:** 10.3390/s22166313

**Published:** 2022-08-22

**Authors:** Emmanuel Utochukwu Ogbodo, Adnan M. Abu-Mahfouz, Anish M. Kurien

**Affiliations:** 1Department of Electrical Engineering, Tshwane University of Technology, Pretoria 0001, South Africa; 2Council for Scientific and Industrial Research (CSIR), Pretoria 0001, South Africa

**Keywords:** 5G, 5G NB-IoT NTN, cryptographic, endogenous security, LPWAN-IoT, non-terrestrial satellite network (NTN), QoE, QoS, smart cities, ubiquitous, LEO satellite, LoRa, LPWAN

## Abstract

Addressing the recent trend of the massive demand for resources and ubiquitous use for all citizens has led to the conceptualization of technologies such as the Internet of Things (IoT) and smart cities. Ubiquitous IoT connectivity can be achieved to serve both urban and underserved remote areas such as rural communities by deploying 5G mobile networks with Low Power Wide Area Network (LPWAN). The current architectures will not offer flexible connectivity to many IoT applications due to high service demand, data exchange, emerging technologies, and security challenges. Hence, this paper explores various architectures that consider a hybrid 5G-LPWAN-IoT and Smart Cities. This includes security challenges as well as endogenous security and solutions in 5G and LPWAN-IoT. The slicing of virtual networks using software-defined network (SDN)/network function virtualization (NFV) based on the different quality of service (QoS) to satisfy different services and quality of experience (QoE) is presented. Also, a strategy that considers the implementation of 5G jointly with Weightless-N (TVWS) technologies to reduce the cell edge interference is considered. Discussions on the need for ubiquity connectivity leveraging 5G and LPWAN-IoT are presented. In addition, future research directions are presented, including a unified 5G network and LPWAN-IoT architecture that will holistically support integration with emerging technologies and endogenous security for improved/secured smart cities and remote areas IoT applications. Finally, the use of LPWAN jointly with low earth orbit (LEO) satellites for ubiquitous IoT connectivity is advocated in this paper.

## 1. Introduction

Considering the quick growth of the population today viz-a-viz industrialization with urbanization, the demand for public resources and satisfactory public services continues to drastically increase. Addressing this recent trend of the huge demand for resources and ubiquitous use for all citizens has led to the conceptualization of technologies such as the Internet of Things (IoT) and the emergence of smart cities. The perception of smart cities functions in a sophisticated urban community by integrating numerous intricate infrastructural systems, human inclusion, technologies, economy, and societal inclusion. Smart cities have been envisaged to provide smartness in managing domains such as transport and mobility, health care, natural resources, electricity and energy, homes and buildings, commerce and retail, society and workplace, industry, agriculture, and the environment. The existing IoT architecture is deficient in suitable communication and security capabilities to adequately support these smart cities applications domains.

The legacy wireless communication such as Wi-Fi, and 3G/4G cannot support ubiquitous massive IoT applications. To fully actualize this high demand for resources, a new global wireless standard such as the 5G network has been envisioned to support emerging applications such as enhanced mobile broadband (eMBB), ultra-large connection, and ultra-low latency, massive machine-type communication (mMTC) for IoT applications. On the other hand, remote areas and rural communities do not have adequate infrastructure to support their agribusiness, broadband access, telemedicine, distance education, etc. Leveraging 5G networks in remote or rural areas will certainly improve these remote areas’ applications.

5G is relatively deployed in mid-bands and millimeter-wave bands. These bands have a medium to short propagation range with low penetration power rate properties, which requires a lot of infrastructures to achieve complete coverage for both urban and remote areas (rural communities). However, ubiquitous IoT connectivity can be achieved to serve both urban and underserved remote areas by deploying a 5G network with Low Power Wide Area Network (LPWAN). This will enable billions of smart devices to interconnect autonomously and support massive machine-type communication (mMTC) services and IoT applications such as connected cars, smart metering, smart homes, smart cities, smart agriculture, smart health care, and sensors [1,2]. The entire scenarios of applications supporting industrialization with urbanization and remote areas will be difficult since applications will require smarter, easier, reliable, faster, and more scalable architecture than the conventional ones. A proper IoT architecture is required to fully support emerging IoT applications and services in terms of security and ubiquity interconnectivity.

Historically, the communication aspect of IoT makes use of short-range communication technologies such as ZigBee, Bluetooth Low Energy (BLE), Wi-Fi, and GSM. Recently, long-range, low-power, and low-cost devices, which are also called LPWAN devices such as LoRa, Sigfox, LoRaWAN, NB-IoT, etc., are now used for IoT connectivity [3]. Due to high service demand and data exchange, the current architectures will not offer flexible connectivity to many IoT applications.

5G, on the other hand, has been envisaged to support emerging technologies. However, due to infrastructural cost constraints, 5G alone cannot provide ubiquitous coverage. However, it can enable LPWAN to achieve wider coverage and support for massive machine-to-machine (M2M) services/IoT applications. For example, a 5G mobile network can serve as backhaul connectivity to LPWAN’s gateways/base stations (BS).

LPWAN is a low power, long-distance range, low energy consumption, and low complexity technology. Expensive infrastructure requirements will be reduced when using LPWAN. This is due to its long-range combined with a star/mesh topology. In addition, spectrum cost will be reduced when LPWAN operates in an optimized licensed spectrum (i.e., a private 5G network with a shared license spectrum) and there is no cost for the unlicensed spectrum. Hence, integrating 5G networks with LPWAN technologies based on novel architecture and security and privacy measures will go a long way to improve smart cities and remote areas applications. Further, poor architecture is deficient in reliable inter-communication protocols which can pose challenges such as radio frequency (RF) and cell edge interferences [4], security and vulnerabilities issues [5], limitation in the support for ubiquity connectivity, and lack of adequate support for emerging technologies. These emerging technologies include Network Function virtualization (NFV); Software Defined Networks (SDN); Network slicing; Advance Interference Mitigation Strategy (AIMS); Mobile Edge Computing (MEC); Mobile Cloud Computing (MCC); Advanced security mechanisms such as 5G Endogenous security; Data/Big Data Analytics; Machine learning/Artificial intelligence algorithm; and new communication technologies such as 5G New Radio, LPWAN, and Heterogeneous network. Consequently, a new state-of-the-art architecture based on emerging technologies is vital to address the stipulated challenges.

This paper is inspired by the potential and upcoming requirements of the 5G and LPWAN-IoT architecture to enable billions of IoT devices for improved and secured smart cities and remote area applications. This paper surveys LPWAN-IoT for improved smart cities by considering the following factors in IoT: 5G and LPWAN Integration, Endogenous Security, LEO Satellite, Cell-edge interference, and 5G-based IoT (5G eMTC, 5G NB2-IoT-enhanced, and new radio (NR) reduced capability (RedCap) IoT device).

The 5G-based IoT is an enhancement of the existing 4G-based IoT (LTE-M and NB1-IoT). This was recently finalized by the third-generation partnership project (3GPP) in the latest concluded 5G Release-17 [6,7].

The contributions of this paper are as follows: Categorize smart cities’ applications based on the finding from the exploration and investigation of 5G and LPWAN-IoT with respect to emerging technologies;Identify research gaps in the aspect of security based on intensive investigation of security challenges in 5G and LPWAN-IoT. Research gaps such as the need of applying endogenous and cryptographic security in 5G and LPWAN-IoT are unarguable;Advocate the remediation of cell-edge interference problem using a special television white space (TVWS) strategy as backhaul connectivity for LPWAN-IoT solutions in a smart city scenario;Determine adequate strategy based on emerging technology for joint consideration of the quality of service (QoS)/quality of experience (QoE) in different application requirements and varying end users in 5G and LPWAN-IoT. In this respect, QoS and QoE are considered jointly in an IoT application;Identify optimal technologies that will address the ubiquity connectivity bottleneck in the LPWAN-IoT ecosystem. For example, the use of non-terrestrial networks (NTNs) such as low-earth orbit (LEO) satellite constellations integration in LPWAN-IoT is advocated.

The rest of this paper is organized as follows: In Section 2, related works in literature are discussed. In Section 3, various architectures involving 5G, LPWAN-IoT, and Smart Cities are explored and investigated. In Section 4, security, and vulnerability challenges, including endogenous security, are discussed. In Section 5, further challenges and solutions are presented. Future research directions are discussed in Section 6. Finally, Section 7 concludes the paper.

## 2. Related Survey Papers

Previous surveys and research studies that consider 5G, IoT/LPWAN, and smart cities have been carried out from different perspectives. Shancang et al. [8] surveyed the current research state-of-the-art of 5G IoT, key enabling technologies, and main research trends and challenges in 5G IoT. Jesus et al. [9] presented a comprehensive review and analysis of research works proposing security solutions for the 5G-LPWAN integration. In [10], the state-of-the-art of IoT application requirements along with their associated communication technologies is explored. In [11], the authors investigated technical issues including a review of recent advances and machine learning-assisted solutions in mMTC. Mekki et al. [12] provided a comprehensive and comparative study of LPWAN technologies which serve as efficient solutions to connect smart, autonomous, and heterogeneous devices for large-scale IoT deployment. Fujdiak et al. [13] provided a detailed discussion of the potential security threats, features, and mechanisms for LPWAN. Their work focuses on the security aspects related to the use of LPWAN and IoT in 5G.

The authors in [14] analyzed the current state of the art of the existing security and privacy solutions tailored to 5G, including emerging paradigms, such as IoT, fog computing, and blockchain. In [15], typical security and privacy issues are identified in 5G. Then, potential solutions to secure 5G networks from several perspectives, including the overall 5G security framework and IoT are provided. Zhang et al. [16] presented a comprehensive detail on the core and enabling technologies that are used to build the 5G security model; network softwarization security, Physical layer security, and 5G privacy concerns, among others.

In [17], an overview of the network architecture and security functionality of the 3GPP 5G networks was presented. In addition, the work focuses on the new features and techniques including the support of massive IoT devices, Device to Device (D2D) communication, Vehicle-to-Everything (V2X) communication, and network slicing which contribute to huge challenges for the security aspects in 3GPP 5G networks. In [18], a comprehensive review of emerging and enabling technologies related to the 5G system that enables IoT, including a review of LPWANs, security challenges, and its control measure in the 5G IoT scenario is provided. Bembe et al. [19] provided a comprehensive study of the current state-of-the-art of LPWAN suitable to meet the requirements of IoT, while uniquely providing LPWAN’s modeling techniques, performance metrics, and their associated enablers.

In [20], the review of the current trends in LPWAN technology with an emphasis on the services it provides and the challenges it faces, including the industrial paradigms for LPWAN implementation are presented. Bocker et al. [21] analyzed the capability of LoRaWAN as a complementary solution in unlicensed frequency bands to contribute to given 5G requirements for specific mMTC applications in large-scale deployments. Malaram et al. [22] presented an overview of the key enabling technologies including emerging technologies for 5G IoT applications. In [23], the authors reviewed various security threats and vulnerabilities at the Physical (PHY) layer, Media access (MAC) layer, and Network layer. Layer-wise integrated approach solutions are provided to mitigate such attacks in IoT. Hassan et al. [24] reviewed the mobility management solutions in LPWAN networks and investigated how they ensure security. The basic IoT security requirements and the typical IoT protocol stack, including the existing mobility management solutions in LPWAN were presented.

Furthermore, the review work in [25] focuses on defining a systematic and powerful approach to identifying the key characteristics of LPWAN-IoT applications, including their requirements and the associated design considerations. Xinsheng et al. [26] analyze the security requirements of 5G business applications, network architecture, the air interface, and user privacy. The development trends of 5G security architecture with a focus on endogenous defense architecture which represents a new trend in 5G security development are presented. Ahmad et al. [27] provide an overview of the security challenges in clouds, SDN, NFV, and user privacy, including solutions to these challenges in 5G networks. Xiaowei et al. [28] described the basic new concepts in the 5G core network architecture and its security implications based on the Third Generation Partnership Project (3GPP), including an overview of the two services which are Vehicle to Everything (V2X) and IoT. Rahimi et al. [29] proposed a next-generation IoT architecture based on new technologies in which the requirements for future applications, services, and generated data are addressed.

In general, considering the various surveys conducted, it is obvious that the surveys and studies that consider 5G and IoT/LPWAN have been conducted in one or some of the following contexts such as enabling/communication technologies, protocols, applications, architecture, interoperability, standardization, challenges, security, and privacy. None holistically considers cell edge interference, QoE in application requirements, ubiquity connectivity covering smart cities/remote areas applications, and architecture with the advanced security mechanism. However, this work holistically considers the aforementioned architecture possessing advanced endogenous security and ubiquity connectivity for improved smart cities and remote areas applications in the LPWAN-IoT. Also, it considers 5G jointly with LPWAN-IoT with an extension to a unified 5G and LPWAN-IoT architecture that will holistically support integration with emerging technologies and endogenous security for improved/secured smart cities and remote areas applications. Also, hardening (i.e., applying advanced security mechanism) of security standard (CIAA) including lightweight asymmetric cryptographic encryption for LPWAN to mitigate security breaches is advocated in this paper. Overall, the use of LPWAN jointly with LEO satellites for ubiquitous IoT connectivity is supported in this work.

Table 1 shows the existing surveys in consideration of the integration of the highlighted factors in IoT. The existing surveys on LPWAN-IoT consider a few of the highlighted factors. However, none considers the 5G-based IoT (5G eMTC, 5G NB2-IoT-enhanced, and new radio (NR) reduced capability (RedCap) device). Also, no survey considers cell edge interference and the endogenous security for LPWAN-IoT as seen in Table 1. In addition, none of the surveys considers the grouping of smart city applications including remote area applications concerning their technical requirements. Though some of the surveys consider QoS, they did not consider the quality of experience (QoE). Equally, there was no joint consideration of QoS and QoE in an LPWAN-IoT application. Likewise, some of the surveys did not consider cryptographic security. Hence, it is obvious that some of the key aspects are not considered by several surveys. However, the consideration of those omitted factors depicts the novelty approach in this paper. Details of this paper’s consideration of the highlighted factors in Table 1 are provided in the subsequent sections.

## 3. Architectures in the 5G and LPWAN-IoT and Smart Cities

In this section, the architectures in the 5G and LPWAN-IoT and Smart Cities are elucidated.

### 3.1. 5G Network Architecture

Currently, 4G/LTE networks are not able to support the mMTC adequately. 5G networks are poised to provide the speediest cellular network data throughput with very low latency/delay and support for ultra-dense network connections. Recently, numerous works on 5G networks for IoT have been investigated [1]. The 5G network architecture is the foundation that will enable IoT applications. Compared with 4G/LTE cellular networks, the 5G core network, radio access network (RAN)/C-RAN (cloud-RAN), and the cloud-based data/5G new technologies must thoroughly be redesigned to provide massive connectivity for large varying IoT applications. The 5G network architecture is illustrated in Figure 1. When compared to existing cellular (4G/LTE), the 5G networks can provide applications users with faster speeds of up to 10 Gbps while maintaining reliable connections of up to multiple thousands of devices at the same time [30,31,32].

The following are the components descriptions of the 5G network:NG-RAN: This is the next generation radio access network which comprises gNB and ng-eNB;gNB: Serves as base station and provides access to a 5G UE (user equipment) over a 5G NR (New Radio) air interface. Also, gNB connects to 5G Core, as well as to 4G evolved packet core (EPC);ng-eNB: Connects to 5G Core and serves as base station by providing access to a 5G UE over 4G radio;5G NR: 5G New Radio brings performance, flexibility, scalability, and efficiency to spectrum usage. Spectrum bands include low-band (<1 GHz), mid-band (3–6 GHz), high-band (24–86 GHz), and ultra-high band (millimeter-wave band (30–300 GHz));5G Core (5GC): Leverages a service-based architecture comprising many interconnected Network Functions (NFs);5GC control plane: This includes Access and Mobility Management Function (AMF) and Session Management Function (SMF);5G user plane: This includes User Plane Function (UPF);5GC Control and User Plane Separation (CUPS): This allows the centralized control plane functions while distributing user plane functions nearer to users for better performance;NG: This is the interface between base stations and 5G network functions servers;xn: This is the interface between base stations.

Current challenges associated with 5G Architecture include backhaul design issues, QoS/QoE issues, emerging technologies integration issues, interference management issues, security and privacy issues, and ubiquitous connectivity issues. Hence, a reliable 5G network architecture should be able to address the stated challenges. For example, the backhaul design issues can be addressed by implementing a 5G architecture that supports diversified backhaul connectivity, that is, backhaul comprising a 5G low-band, mid-band, and high-band deployed at respective locations based on application needs. This is indeed a cost-effective architecture deployment solution; whereby backhaul resources are utilized where necessary.

The emerging technologies integration issues can be addressed by proper integration of these emerging technologies such as Network Function virtualization (NFV), Software Defined Networks (SDN), Network slicing, Machine learning (ML)/Artificial intelligence (AI) in the 5G architecture in order to satisfy the various use cases of 5G network solution demand. This will help to improve the QoS/QoE of the network. Further discussion on this aspect is provided in Section 5.

The interference management issues including cell edge interference especially in a dense network environment can be tackled with the use of multiple-input-multiple-output (MIMO) antennas in 5G architecture. This will help to offload some of the connections to the available massive MIMO antennas. Also, the cell edge interference problem can be overcome by implementing 5G jointly with a very long-range LPWAN. Detailed information on this is provided in Section 5.

A reliable 5G architecture should have adequate support for security and privacy solutions. A key security concern is that of the physical (PHY) layer security, that is, security involving the device, channel, and over-the-air (OTA) interface. One of the measures to circumvent this PHY layer security is the integration of endogenous security in the 5G architecture. Further discussion on this is provided in Section 4.

The problem of ubiquitous connectivity can be addressed by adequate integration of LEO satellite constellations jointly with LPWANs in the 5G architecture. This will help for global coverage of 5G and LPWAN-IoT connectivity. The next Section gives detailed information on this.

### 3.2. Ubiquitous Connectivity Challenges and Solutions in 5G and LPWAN-IoT

Over 80% of the earth is not covered by broadband services. Thus, greater proportions of the earth should be covered by broadband services for the ubiquitous Internet of Things (IoT) to be actualized. One of the key technologies that will support ubiquitous IoT connectivity is the Low Earth Orbit (LEO) satellites. The fifth generation (5G) network will also support the ubiquitous IoT. This is because 5G has been envisioned to enable massive IoT. In the past, broadband services and IoT applications such as smart metering, smart home, smart building, smart cities, e-health, factory automation, asset tracking, and so on have been supported by communication systems such as cellular (2G, 3G, GSM, and 4G), wireless fidelity (WI-FI), Bluetooth, and ZigBee. Presently, IoT applications are being supported by low power wide area networks (LPWAN) such as LoRa, Sigfox, LTE-M, and NB-IoT for long-range connectivity solutions. Yet, historically, the connectivity solutions have not addressed the challenge of ubiquitous IoT. More recently, the LEO satellite has emerged as a potential technology for ubiquitous IoT connectivity to serve the underserved area including ocean, forest, and remote areas, especially in developing countries region. Due to the closeness of the LEO satellite to the earth, it has an appreciable data rate and low latency better than the geosynchronous equatorial orbit (GEO) satellite. To integrate non-terrestrial networks (NTNs) such as LEO satellite constellations with terrestrial networks (TNs), 5G NB-IoT standards for Non-Terrestrial Networks (NTNs) have been standardized in the 3GPP release-17’s 5G NB-IoT NTN specifications [33]. This will help to facilitate connectivity between IoT devices via TNs and NTNs globally. Hence, LPWAN-IoT can leverage 5G NB-IoT NTN for ubiquitous connectivity in smart city applications.

### 3.3. IoT Architecture

Today’s IoT applications are geared towards enabling a smarter livelihood for everyone which involves the interconnection among several smart devices for smart cities. The adequate architecture will enable reliable interconnections of various smart devices in our homes, buildings, cities, and industries for the industrial internet of things (IIoT) [34]. Hence, architecture for IoT has been viewed from different perspectives by various IoT designers since there is no specific universally agreed IoT architecture. Different architectures have been advocated by numerous researchers. A study of available IoT architectures is provided in [35,36,37]. The general IoT architecture is the layered approach architecture. Its simplest or basic form is a three-layer architecture as shown in Figure 2. This architecture consists of the following [38,39]:Perception Layer (Layer 1): This is the layer that considers physical devices and sensors that collect data/information about the entity as well as the environment;Network Layer (Layer 2): This is where the connections to other things, devices, and services, including the processing of sensor data, are carried out;Application Layer (Layer 3): This layer defines the applications where the IoT is to be deployed, including provisioning application services to the users.

The three-layer architecture is meant only for the basic requirements of IoT. Sophisticated applications and business solutions require more layers in the IoT architecture. For instance, there are four-layer, five-layer, six-layer, and seven-layer architecture. Considering the present-day advanced smart applications due to the emerging technologies and 5G network support, a seven-layer IoT architecture is preferred for many IoT applications. An illustrative of the seven-layer IoT architecture is illustrated in Figure 3.

### 3.4. LPWAN Architecture

LPWAN was developed to allow long-range communications at a low data rate and with low energy consumption. Hence, LPWAN devices are considered to be low complexity long-range devices. LPWAN works in the license-free frequency bands and the licensed frequency (cellular) bands. The LPWAN technology use case is not very common though it is gradually increasing and involves numerous different technologies due to different manufacturers/vendors. A typical LPWAN architecture is shown in Figure 4, which is common to many LPWAN technologies.

The basic network topology for the LPWAN deployment is the star topology. Other topologies such as the tree topology are used with a greater number of nodes to increase the range. To increase the range further and with several nodes, a mesh topology is used. However, the mesh topology has higher complexity and energy consumption with minimal delay. Detailed discussions on the network topologies including their merits and demerits can be found in [40,41,42].

#### Technologies Concepts for LPWAN Market Players

The conceptual technologies for the various LPWAN market players are discussed as follows:LoRaWAN

LoRaWAN is a low power wide area network built on LoRa which is envisioned for low-cost, low battery power-operated devices for wider coverage and ubiquity connectivity. LoRaWAN is owned by Semtech Corporation. It is in the family of the LoRa Alliance, an open, non-profit association of industrial vendors for IoT connectivity solutions. The LoRa is a proprietary PHY layer protocol, and the LoRaWAN was developed to define the upper layers protocols. It performs primarily as a network layer protocol for managing communications between LPWAN gateways and end-node devices as a routing protocol, maintained by the LoRa Alliance. It uses sub-1 GHz frequency bands with a proprietary modulation technique based on chirp spread spectrum (CSS) modulation. It covers a range of 10 km., and about 20 km in rural areas. The data rate is from 0.3 kbps to 37.5 kbps. Detailed upper layers specifications of the LoRaWAN are available in [43]. Simple ALOHA is used for channel access in the MAC layer. The use of different channels and orthogonal codes enables LoRa to establish a connection between multiple nodes [40]. LoRaWAN provides a reliable communication system during moderate traffic, though it encounters the following problems such as duty cycle management, frequency hopping, and channel access mechanisms [23] once the traffic exceeds the maximum limits.

Sigfox

Sigfox is a Global French network operator. Its PHY layer protocol is also proprietary and operates in a licensed-free band of sub-1 GHz radio bands. It is suited for low-energy devices and comprises base stations with cognitive-based software-defined radios (SDR) and IP-based network servers. It uses BPSK modulation techniques and ultra-narrow bands (UNB) of 100 Hz for carrier signals and spectral efficiency improvement. It has the following advantages such as improved receiver sensitivity, long-range, reduced power consumption, and high penetration rate through a sole object, which makes it suitable for deployment in underground or rough topography [44]. In rural or open space setting the connectivity, the range is over 40 km. However, only 140 uplink messages with a payload of 12 bytes and up to 4 downlink messages with each a payload of 8 bytes per day can be communicated through Sigfox [41]. Sigfox network reliability can be improved by using a time and frequency variation strategy with multiple transmissions [45].

DASH7

DASH7 is based on the DASH 7 Alliance Protocol (D7AP) and was derived from the ISO 18000-7 standard. It is a low-power, long-range device that also operates in a licensed-free band of sub-1 GHz radio frequency. It uses narrowband two-level GFSK modulation with a channel bandwidth of 25 kHz or 200 kHz along with data whitening and forward error correction features [25]. It has a data rate of 167 Kbps, low latency with nodes mobility of 2 km, range, and uses a low power wake-up system to reduce energy consumption, extending the battery life up to 5 or more years [46,47]. DASH7 architecture consists of endpoint devices, sub-controllers, gateways, and servers. The endpoint devices/nodes follow a strict duty cycle schedule, while sub-controllers collect the data packets from the endpoint nodes with some sleep cycles and low power restrictions. Gateways are always active to collect packets from sub-controllers and endpoints and then send them to the network server [25];

Nwave

Nwave is a solution developed by Nwave incorporation for smart parking systems [48]. It is a UNB technology and operates in the sub-1 GHz licensed-free ISM band. It advocates long-range and high node density compared with Sigfox and LoRa at the expense of higher energy consumption [49]. Nwave supports data rates of up to 100 bps with a range of up to 7 km and 8 years of battery life. It has an appropriately real-time data collection and management software system for monitoring and control;

Weightless

Weightless belongs to the Special Interest Group (SIG). SIG introduced an open standard for three LPWANs connectivity such as Weightless-W, Weightless-N, and Weightless-P [50]. Weightless-W operates in the frequency band of TV whitespaces (TVWS). It uses quadrature amplitude modulation (QAM) and differential binary phase-shift keying (DBPSK) modulation with spreading techniques. The data rate is up to 10 Mbps. Data transmission to the base station is carried out in a narrow band, which reduces energy consumption. Weightless-N also operates in the sub-GHz band and uses DBPSK modulation. It supports one-way communication from the end devices to a base station, hence it is the most energy-efficient Weightless-SIG standard. Though, one-way communication poses some limitations on the Weightless-N operation. Weightless-P also operates in the sub-GHz band. It supports two-way communications with a data rate of 100 kbps and uses Gaussian frequency shift keying (GMSK) and quadrature phase-shift keying (QPSK) modulation techniques for different applications;

Ingenus (RPMA)

Ingenu is a proprietary LPWAN solution and uses a random phase multiple access (RPMA) scheme with flexible spectrum regulations. This flexibility makes RPMA a potential for higher throughput, capacity, scalable, and wider coverage [51]. Ingenu operates in the 2.4 GHz band, though with a high penetration rate, which makes it possible to strive underground and for environmental sensing [51]. It was designed for machine-to-machine (M2M) communications to serve IoT applications in utilities, oil and gas sectors, agriculture, asset tracking, fleet management, smart grids, and smart cities;

Cat 1/Cat M1(LTE-M)

LTE Category 1 (Cat 1) is a standard that was introduced in the third-generation partnership project (3GPP) Release 8 as an introduction to M2M communications [52]. It is the first LTE IoT-specific variant. Compared to its predecessor like LTE Cat 2 or Cat 3, it provides better power efficiency, extended idle and sleep modes, lower complexity, and longer coverage, which makes it easier for massive IoT deployments. Based on these features, it falls under LPWAN technologies and operates on a licensed frequency (LTE cellular) band. It has up to 10 Mbps download and 5 Mbps upload data rate with a channel bandwidth of 20 Mhz. Hence, it is suitable for IoT applications that require higher throughput. However, the newer variant like the Cat M1 or LTE-M in the 3GPP Release 13 LPWAN technology has lower complexity, lower data rate, and longer range. The LTE-M also operates in the licensed LTE spectrum and provides connections for M2M communications. It provides extended coverage than LTE Cat 1 and offers a reliable path towards a 5G enabled M2M IoT solution [12]. It has a low latency of a few seconds, and data rate of up to 1 Mbps as well as low energy consumption with longer battery life. It supports devices with a wide range of message length sizes, including support for device mobility to some extent [11]. It has support for high capacity and scalability;

Narrowband IoT

Like the LTE-M, the Narrowband IoT (NB-IoT) is also in the 3GPP Release 13 LPWAN technology. However, it operates with lower complexity and at a lower data rate when compared to LTE-M. NB-IoT also operates in the licensed LTE band but with the flexibility for deployment in the global system for mobile communications (GSM) licensed frequency bands (700 MHz, 800 MHz, and 900 MHz). It uses two-way communication in which the orthogonal frequency division multiple access (OFDMA) is used for downlink, and single carrier frequency division multiple access (SC-FDMA) is used for uplink [53]. It connects 50,000 devices per base station and uses a channel bandwidth of 180 kHz to establish communication. It has a 200-kbps data rate for downlink and 20 kbps for uplink, with a battery life of up to 10 years. However, an enhancement to the NB-IoT is the 3GPP Release 14 enhanced NB-IoT, it is called LTE Cat NB2, which enhances the NB-IoT protocol in several ways such as high positioning accuracy; introduces NB-IoT Multicast; enhances device mobility; high data rate; increases peak data rates; introduces NB-IoT Multi-carrier operation; additional lower device power class; lower latency and new NB-IoT frequency bands allocation [54]. Further enhancement of the LTE-M and NB2- IoT which were previously on 4G/LTE-enabled platforms has now been finalized as a 5G-based IoT (eMTC and NB2-IoT-enhanced) respectively, this was announced in the latest release of 3GPP Rel-17 [6]. This will support IoT devices at 5G narrowband features with an appreciable data rate than the previous LTE-M and NB-IoT. The latest enhancement in the 5G user equipment (UE) is the New Radio (NR) Light-IoT or 5G reduced capability (RedCap) IoT device which will operate on 5G NR and is part of release 17 of the 3GPP [6]. Table 2 shows LPWAN technologies comparisons, while Table 3 shows 3GPP release numbers and their details.

EC-GSM-IoT

Enhanced coverage-global system for mobile Internet of Things (EC-GSM-IoT), introduced by 3GPP in its release 13 for LPWAN cellular IoT (CIoT), with support for similar coverage and longer battery life as NB-IoT [55]. The design is based on enhanced General Packet Radio Service (eGPRS) for the IoT, and existing GSM Networks can be upgraded using a software application to ensure wider coverage and accelerated time of deployment. Also, optimization strategies are deployed in EC-GSM-IoT for efficient battery life of about 10 years for a wide range of use cases [10]. The channel bandwidth of the EC-GSM-IoT is 200 kHz. It has a peak data rate of 70 kbps and 240 kbps. EC-GSM-IoT provides multi-fold improvement in the coverage for low-rate applications. It also has a high penetration rate such as the ability to penetrate deep indoor basements [56] where multiple smart meters and parking sensors are installed including remote areas sensors deployed for agriculture or asset monitoring use cases [55].

Table 2 below provides a summary of the various LPWAN technologies. Table 3 exemplified the 3GPP release date and associated details;

IEEE 802.11ah

The IEEE 802.11ah is an emerging wireless networking standard, which is also called Wi-Fi HaLow. It operates at the sub 1 GHz unlicensed spectrum bands and provides a wider coverage range up to 1.5 km compared to the 2.4 GHz and 5 GHz Wi-Fi networks. The data rate of up to 30 Mbps is achievable when using 16 MHz channel bandwidth. The Wi-Fi Halow has a low energy consumption, and it can be used for indoor and outdoor applications [57,58]. Hence, it is a low power wide area network (LPWAN) IoT device due to its features. In addition, it can support various IoT applications due to its varying physical layer characteristics such as numerous channel bandwidths (1 Mhz, 2 Mhz, 4 Mhz, 8 Mhz, and 16 Mhz) and modulation and coding schemes up to 256 QAM (quadrature amplitude modulation).

### 3.5. Smart Cities Conceptual Framework

A smart city is an urban area that uses digital technologies to enrich residents’ lives, improve infrastructure, modernize government services, enhance accessibility, drive sustainability, and boost-up economic development [59].

A modern smart city is a framework principally comprised of an intelligent network of connected objects and machines using Information and Communication Technologies (ICT) such as 5G cellular, LPWAN wireless technologies, and the cloud to collect/transmit data, develop, deploy, and promote sustainable development practices to address growing urbanization challenges. It improves infrastructure, efficiency, convenience, and quality of life for residents and visitors likewise [60]. The smart city concept integrates ICT’, and various objects/devices connected to the ‘IoT’ network to optimize the efficiency of city operations and services and connect to citizens [61].

Smart cities are defined as smartness both in how their governments harness technologies as well as in how they monitor, analyze, plan, and govern the city [62]. Hence, smart cities can easily be conceptualized based on the Smart City Wheel, developed by Dr. Boyd Cohen [63,64] in which a smart city is defined as the integration of the following six attributes: Smart People; Smart Government; Smart Environment; Smart Economy; Smart Mobility and Smart Living. Figure 5 shows a clearer illustration of these attributes integrated into the smart city framework.

#### Smart Cities and Remote Area Applications Requirements

Various smart city applications have different technical requirements due to their varying features. Hence, smart city applications can be categorized into three groups based on certain features such as coverage distance, data rate/bandwidth, energy consumption, and latency. Based on these features, technical requirements such as communication networks and devices can be identified for a particular smart city application. For instance, an LPWAN solution can be identified to enable a low data rate with longer distance coverage for smart city applications. The smart cities application groups and remote area applications are:Massive machine-type communication (mMTC) smart cities applications;Critical infrastructure smart cities applications;Enhanced mobile broadband (eMBB) smart cities applications;Remote area applications

Table 4 shows the smart cities applications group with their examples and associated supported features. This smart city application grouping together with LPWAN technologies in Table 2 will help one to know at a glance the LPWAN-IoT device that will support a particular IoT application.

## 4. Security Challenges and Solutions in 5G Network and LPWAN-IoT

This section provides an overview of security in 5G and LPWAN-IoT including security mechanisms and challenges. The respective security challenges and solutions in LPWAN-IoT and 5G are identified respectively. The endogenous security challenges and solutions for 5G and LPWAN-IoT are also discussed.

### 4.1. Overview of Security in 5G Network and LPWAN-IoT

Wireless networks are susceptible to security breaches and vulnerabilities. The security breaches in 5G networks will be greater than the security and privacy issues in the previous wireless cellular generations. Also, because 5G will enable other new wireless technologies, there exist some potential security problems in these new network solutions as well. Hence, it is vital to highlight the security challenges that cut across the 5G ecosystem and the emerging technologies solution. LPWANs solutions together with 5G networks pose varying security challenges. For instance, implementing security mechanisms to mitigate vulnerabilities in cellular LPWAN such as NB-IoT and LTE-M is simplified because NB-IoT and LTE-M are using the 5G native security protocol mechanisms to secure the IoT platform. On the other hand, the non-cellular LPWANs such as LoRaWAN, Sigfox, etc., adjust or modify their security mechanisms to coexist and work with the security mechanism of 5G networks. However, the non-cellular LPWANs solutions have intrinsic weak potential to function with standard security protocol mechanisms such as transport layer security (TLS) and other advanced authentication security. This makes it vulnerable for attackers to easily attack the edge, application server, and the core network. In the first place, LPWAN-based technologies are highly restricted considering the number of messages allowed for transmissions per day and their length. For that reason, considering typical authentication protocols employed in non-restricted systems is not a valid approach for these systems as they make use of several big-sized messages. Also, the LPWAN devices have low computational complexity. This makes it difficult to withstand the complex security overhead required by the 5G operation.

### 4.2. Security Challenges and Solutions in LPWAN-IoT and 5G

#### 4.2.1. Security Challenges and Solutions in LPWAN-IoT

Confidentiality, integrity, authenticity, and availability (CIA^2^) are the essential part of security requirements in information assets and LPWAN-IoT. Confidentiality ensures that only authorized users can have access to data/classified information which cannot be snooped on by unauthorized users. Integrity ensures the ability to safeguard the data/classified information from any intrusion throughout the communication process. Authentication ensures that the device transferring data and the data being transferred is legitimate. Availability ensures that information and network resources are continuously available for legitimate users when needed. CIAA can be implemented in PHY, MAC, and the network layer. Various security mechanisms including the physical layer security (PLS) mechanism are used to avoid data/security breaches by an eavesdrop-attacker. Incorporating cryptographic algorithms used in the MAC offers an advanced level of security. Security mechanisms at the network layer also of some level of security [23].

Security threats occur due to the exposure of the devices connected to the internet through the LPWAN wireless air interface [5]. Security breaches and vulnerabilities in LPWAN-IoT for smart cities and remote areas applications are high because of the massive MTC connectivity. Encryption of the application interface and network request access is essential to avoid possible intrusions. Over-the-air (OTA) security is a key facility that ensures devices are not exposed to security risks over a lengthy duration [65]. OTA can be implemented as an alternative to the authentication method. It is costly to deploy OTA security in non-cellular LPWAN technologies because of the heterogeneity with other user equipment (UE)/devices. Strong encryption and authentication mechanisms such as advanced encryption standards AES) are used in many networks for secure data transmission. However, AES is difficult to implement in some LPWAN solutions due to the low complexity of LPWAN technology. Other security mechanisms which are based on cryptographic standard protocols include Diffie–Hellman (DH) which uses key exchange/management, and Rivest–Shamir–Adleman (RSA) which is used to authenticate digital signatures/key transport. Also, cellular LPWAN uses the subscription identity module (SIM)-based authentication schemes to provide security protection

Some of the major security challenges common to LPWAN-IoT are enumerated below:Flash network traffic: It is expected that the number of end-user devices which will be supported by LPWAN will grow exponentially in 5G due to massive MTC. This will cause substantial changes in the network traffic patterns, thus giving room to security loopholes and malicious activities [13];Jamming attack: This is one of the main problems for LPWAN-IoT. Malicious entities can send a powerful radio signal in the same transmit power as the legitimate application devices and interrupt the radio transmissions [5]. Such jamming signals can adversely affect LPWAN transmission. For instance, concurrent LoRa or Sigfox transmissions at the same frequency and spreading factor can interfere with each other;DoS attacks: DoS and DDoS (distributed denial of service) attacks can exhaust computing and network resources such as energy, storage, and connectivity [66]. This will prevent legitimate users from gaining access. Intermittent requests or explicitly crafted requests generated toward the legitimate network in enormous numbers by illegal LPWAN users can mar the normal operation of the network or bring the network to a halt;Security of radio interface keys: In previous wireless network generations, including 4G, the radio interface encryption keys are generated in the home network and transmitted to the visited network over insecure links that exposes the keys [66];Compromising devices and Network keys: Devices including the microcontroller unit (MCU) can be compromised. For instance, in a particular security breach in LoRa, the Signal mousetrap was used as a target device. The hardware unit was altered to expose the UART serial lines between the MCU and the LoRa radio module. A regular FTDI chip was connected to the serial line to interrupt and capture all the transactions between them. Whenever the mousetrap was reset, the host MCU issued commands to configure the network keys of the radio module. Using these keys and a custom LoRa device, a LoRa mouse trap can be impersonated to send data as if it were coming from the mouse trap [5].

The following security mechanisms or countermeasures can be implemented in LPWAN-IoT to mitigate the security challenges and vulnerabilities risks. Detailed illustrations are provided in [13,67,68,69,70,71,72]:Secured credential provisioningAuthentication (device, network, message, and subscriber).Data confidentialityVirtual Private Network (VPN) securityE2M (end to the middle) security and E2E (end-to-end) securityReplay protection.Reliable deliveryPrioritization.Updatability.Network monitoring and filteringAlgorithm negotiationClass break resistance.Certified equipmentSecured IP network

Table 5 shows various security threats and their applicable countermeasures in LPWAN-IoT. The concerned security requirements are the security standards that can be breached by an attacker to launch an attack. Therefore, hardening the security standards or requirements will eliminate security breaches and vulnerabilities, thereby making it difficult for an attacker.

The descriptions of the common security threats are provided below:Replay attack: A replay attack is a playback attack in a network in which valid data transmission is maliciously or fraudulently repeated or delayed. It is carried out either by the originator or by an attacker who intercepts the data. The countermeasure to avoid this attack is replay protection such as random session key establishment used, which will be valid for single interaction between sender and receiver. Also, timestamps that come with a time limit can be used, attacker won’t be able to send a message with expired timestamps. In addition, a one-time password (OTP) can be used every time a session is established, or data is sent across;Spoofing attack: A spoofing attack is when an attacker masquerades or impersonates another device or user on a network to launch attacks against network hosts, steal data, spread malware, or bypass access controls. Strong authentication can be used to avoid or mitigate spoofing attacks;DoS attack: Denial of service attack (DoS) is a threat in which an attacker floods a network with malicious data traffic to exhaust network resources, thereby, making the network services unavailable to the legitimate user. DoS can be avoided using firewalls, network monitoring, and filtering;Wormhole: An attacker forms a tunnel between two or more compromised nodes so that all the traffic is transmitted through it. The cybercriminal aims to modify the logical topology of the network location to impede network traffic. End-to-end (E2E) security and encryption can be used to avoid this attack;Signal jamming: An attacker deliberately jams or blocks the signal of legitimate wireless communications. This can be avoided by secured credential provisioning;Eavesdropping: It is the act of secretly or sneakily listening to the private communications of others without their consent to gather information. This can be mitigated by strong data confidentiality;Man in the middle attack: An attacker secretly relays or alters the communications between two entities that are directly communicating with each other, thereby, intercepting the message with possible modification. This can be avoided using a reliable delivery mechanism;Floods attack: This is like a denial-of-service attack (DoS) and can be mitigated using firewalls, network monitoring, and filtering.Session hijacking: This is a method an attacker uses to take control of another user’s session and gain illegal access to data or resources. Certified equipment and encryption can be used to mitigate this threat;Injections attack: An attacker provides malicious code to an application and changes the operation of the application by manipulating it to perform certain commands. This can lead to a DoS attack and can be avoided with firewall, network monitoring, and filtering;Sniffing attack: An attacker intercepts data by capturing the network traffic using a packet sniffer tool, then analyzes the network to gain information and eventually disrupt the network. This can be mitigated by using firewalls, network monitoring, and filtering;Sybil attack: It is a type of attack in which an attacker disrupts the service’s reputation system by creating many pseudonymous or false identities and uses them to gain extreme access. This can produce wrong reports and loss of privacy. Updating the application and password authentication can mitigate this threat;Sinkhole attack: It is a type of attack in which a compromised node attracts network traffic by advertising its false routing update. This can lead to DoS and can be avoided by E2E security and network filtering.

#### 4.2.2. Security Challenges and Solutions in 5G

Consequent to the 5G security challenges highlighted involving mobile/virtual operations, telecommunications fraud, the IoT, the Internet of Vehicles (IoV), and the SDN, the 5G Public-Private Partnership (5GPPP) Security Working Group introduced to research on security architecture, access control, privacy protection, trust models, security monitoring and management, network slicing security isolation, including other aspects [73]. The next-generation mobile network (NGMN) advocates for user authentication, user privacy protection, and network security, including other aspects [66]. Hence, 5G requires a new security mechanism for new applications, new network architecture, and new air interface technologies. Security is required to authenticate massive device-to-device communications, provide high availability, and support low latency, and low energy consumption for IoT applications.

The advent of new technologies such as SDN/NFV, MEC, and other new technologies introduced some changes and security risks. Yet, the existing 4G/LTE-A security architecture and security key technologies are unable to address these new security problems. Thus, these security issues caused by enormous application scenarios and new technologies pose new challenges for the design of 5G security architecture. The 5G security architecture has been envisioned to support numerous application scenarios as well as a unified authentication mechanism and network slice security including user privacy protection. The 3GPP Working Group-SA3 is responsible for the design of 5G security architecture. The SA3 has enacted that 5G security architecture design and key hierarchy should encompass the basic form [74] as depicted in Figure 6. The simplified 5G architecture of the 5G System involving only the security-related functions is shown in Figure 6. The Abbreviation gives the descriptions of terms including the 5G key hierarchy abbreviations.

Generally, a 5G System consists of the radio access network (RAN) and the core network. The RAN comprises the Next Generation Node B (gNB) which serves as the 5G base station. The core network contains the functions for the management/delivery of the diverse services to the UE or device. The core functions include the authentication server function (AUSF), the access and mobility function (AMF), and the unified data manage- ment (UDM) function which stores the user profiles. 5G network security features can be categorized into two groups. In the first group, all the features secure the communication between the UE and the network OTA interface to the base station.

The other group is meant for features that secure the communication between the different network functions such as between the RAN and the core network, which are the backhaul network interfaces. Also, between the UE and the network, security is provided at two levels or strata. The first level is the access stratum (AS) that protects the control plane (CP) and user plane (UP) between the UE and the gNB transmitted over the packet data convergence protocol (PDCP). The second level is the non-access stratum (NAS) that protects the CP between the UE and the core network transmitted over the NAS protocol.

A run of the primary authentication (PA) procedure is required for mutual authentication between the UE and the network to be successfully carried out [76]. 5G supports two sets of this authentication. The first one is an enhanced version of the authentication and key agreement procedure (AKA) or 5G-AKA, which was developed to support a procedure called the generic bootstrapping architecture (GBA) [72] in the earlier network release. The second one is an EAP-based procedure called EAP-AKA [77] and was developed to support LTE authentication of UEs over the non-cellular type of access networks such as WLAN, and LPWAN. This second set of authentications is the secondary authentication procedure. 5G dictates the use of different session keys for specific protocols and purposes between the UE and the network components. The keys are organized in a hierarchy as shown in Figure 6. At the root of the hierarchy is a key that is shared between the UDM in the home network and the UE where it is securely stored in a smart card. This approach is the key hierarchy and was considered important to meet the strict requirements for isolation and key separation. Mobility of the UE introduces mobility of the security anchor points within the network, which entails a change of the gNB or the AMF serving the UE.

Thus, it is important to follow the principle of categorization so that a ruined key in one network entity does not spread to the other entities. The PA is based on the root key. The other keys are afterward derived from keys higher in the hierarchy for other dedicated procedures. Each key in the hierarchy is shared between the UE and a specific entity called to function in the network. For instance, the K_AUSF_ key is shared with the AUSF, the K_SEAF_ and K_AMF_ keys are shared with the AMF, and the K_gNB_ key is shared with the gNB. The 5G specifications define a particular procedure for the establishment of each key in the hierarchy. For instance, the K_AUSF_ and K_SEAF_ keys are established by the PA procedure which runs between the UE and the AUSF. While the K_AUSF_ key remains in the AUSF, the K_SEAF_ is sent to the target AMF serving the UE and afterward used for the derivation of the K_AMF_ key. The Kg_NB_ is initially established by a combination of procedures involving the AMF, the gNB, and the UE. The UE and AMF use the KAMF to agree on a KGB. The AMF then provides this key to the gNB, in which the UE is connected to the network and finally activates the security OTA between the UE and the gNB based on the K_gNB_ [74,75].

A new feature called Authentication and Key Management for Applications (AKMA) framework was developed by SA3 as an enhancement of the GBA feature that was meant for earlier networks (4G/LTE-A) [74,78,79]. The aim is to utilize an operator authentication infrastructure to bootstrap security between the UE and an application function (AF). Since the UE has a subscription already to access the network and shares security keys with a given operator. Such keys can be used to establish a secure channel for other purposes as well, to secure communication with an application service provider. For example, banks, institutions, tax offices, social security services, and so on. The 5G network system in the AKMA framework supports a UE to be registered in and or attached to both over 3GPP or non-3GPP network access. But the GBA lacks this capability feature for supporting authentication in 3GPP together with a non-cellular network. Further enhancement of the 5G authentication mechanism led to the development of integrated access backhaul (IAB) by the RAN groups in 3GPP SA3 [80,81]. The IAB feature is envisioned to improve the coverage and boost the performance over 5G New Radio (NR) technology.

Moreover, to enable mechanisms for differentiation protection between the CP and UP in 5G security, and support the protection for both slices and applications security, including data security protection, a unified authentication framework for 5G security architecture was proposed by the China IMT-2020 (5G) Promotion Group [26]. This entails UE, access network, serving network, home environment, and service applications. This 5G security architecture contains eight major domains, (see Figure 7). This is illustrated as follows: (1) network access security: security of user data should be guaranteed and this includes confidentiality and integrity of signaling in both the access network and the core network, and the UE and network in CP)/UP; (2) network domain security: the security of exchange in both signaling and user data between different network entities including RAN and service network public or external nodes, home environment and service network external nodes, service network external nodes and network slices, and home environment and network slices; (3) initial authentication and key management: various mechanisms for authentication and key management should be included that exemplify the unified authentication framework, including operator-security-credentials based security credential authentication between UE and 3GPP networks, including the key management of user data protection after successful authentication. Detailed illustrations involving domains (4) to (8) are found in [26].

Table 6 shows the cryptographic mechanisms and their encryption algorithms in 5G-LPWAN-IoT. From Table 6, 5G and most LPWAN use a symmetric encryption algorithm that uses s single key for encryption and decryption of cipher data/message. This is not very secure like the asymmetric encryption that uses public and private keys for encryption/decryption. Most LPWAN use symmetric encryption because of their less computational complexity. The asymmetric encryption such as Rivest-Shamir-Adleman (RSA) and Elliptic Curve Cryptography (ECC) are applicable mainly in 5G and most 3GPP (cellular) LPWANs. The computational complexity in most non-cellular LPWANs cannot withstand the RSA/ECC algorithms. Hence, this is a possible area of research for most LPWANs.

### 4.3. Endogenous Security Challenges and Solution in 5G and LPWAN-IoT

Wireless signals use electromagnetic waves as the carrier to transmit information openly in space. However, the propagation of wireless signals also exposes the endogenous” gene” defects of electromagnetic waves, such that anyone within the signal coverage area can eavesdrop or attack the physical layer [82]. OTA of the physical layer in 5G and LPWAN for IoT solutions are also exposed to endogenous security problems. Traditionally, the existing security mechanisms primarily follow the encryption mechanism in communication, which cannot withstand security issues caused by the openness in wireless channels. Various encryption mechanisms for wireless networks have earlier been cracked. For example, the KASUMI encryption algorithm in 3G was cracked due to loopholes [83]. The existing security mechanisms are being improved to address the past security issues. For example, 4G uses Snow3G/AES/Zuc-based hierarchical keys to mitigate security attacks such as SS7 signaling hijacking in 3G. But it was difficult to deal with the unknown risks and security breaches in the present system. Therefore, it is paramount to address endogenous security problems. This can be achieved by the study of new security mechanisms based on the essential attributes of wireless communication security to deal with known and unknown security threats in wireless systems [82].

Consequently, endogenous security theory can be used to resist both known and unknown security threats. Recently, this theory has made a significant breakthrough as an upcoming security research area in technical advancement and system development [84]. Hence, the research on wireless endogenous security is in the gradual stage. Wireless channel naturally has endogenous security attributes that are dynamic, heterogeneous, and redundant. Based on this, a distinctive endogenous security structure of wireless communication systems can be designed to safeguard the security and reliability of data transmission from the transmitter to receiver. Wireless endogenous security technology has been advocated for the protection of the confidentiality, reliability, and integrity of information in a few recent research. For instance, a wireless channel was used as an executor to realize a secured transmission technology [81] based on random signal scrambling [85,86,87]. Physical layer key generation technology based on a “one-time pad” [88,89,90] is also realized. The realization of the secured communication was brought about by the endogenous security strategy at the physical layer transmission, which enables only users on the legal channel to correctly demodulate the signal, while the signals on the channels in other locations are scrambled and unrecoverable. This endogenous security mechanism can be implemented on LPWAN-IoT for secured smart city applications.

Endogenous security deployment framework in 5G security architecture is a current research focus of the industry. The current development trend in 5G network architecture involves features like clouds, SDN, and virtualization. Based on this, it is necessary to leverage endogenous security strategy regarding OTA networks for new defense mechanisms. Hence, involving the following: physical layer security, lightweight encryption, network slice security, mimic defense, user privacy protection, blockchain, and emerging technologies within the 5G network will lead to 5G endogenous security architecture as shown in Figure 8. This will help to realize a high-confidence, integrated technology solution that defends against both known and unknown security risks [82].

## 5. Further Challenges and Solutions in 5G and LPWAN-IoT

Apart from the security challenges earlier discussed, other challenges and solutions are presented in the following section.

### 5.1. QoS/QoE Challenges and Solutions in 5G and LPWAN-IoT

Quality of Service (QoS) takes into account features such as throughput, latency, etc., of a telecommunications service that bears on its ability to satisfy the stated and implied needs of the user of the service. The International Telecommunication Union (ITU) defines Quality of Experience (QoE) as “the degree of delight or annoyance of the user of an application or service” [91]. QoS involves objectives and thresholds based on technical needs for applications. In QoS, Applications will become much more adaptive to network conditions than today by machine learning (ML) and artificial intelligence (AI) techniques. QoE is based on Experience, i.e., experience on Expectation. In QoE, Expectations will change with new and evolving applications over time [92].

5G is meant to virtually connect everyone and everything with different services in different industries. But different services in different industries require different Quality of Service (QoS)/Quality of Experience (QoE), should we use one physical network or multiple physical networks to satisfy requirements? The answer is one physical network [93]. Hence, 5G can be implemented with a new technology such as SDN/NFV to provide cost-effective and flexible service for the different industries with varying end-user QoS/QoE. This will help to determine and prioritize the instantaneous level of services in the network for prime performance. Network slicing which involves the slicing of virtual networks based on different QoS can be used to satisfy different services. The 5G network slicing is based on SDN/NFV. Consequently, research involving network slicing will help to solve the problem of QoS/QoE encountered by the users. Some works related to QoS and/ or QoE in 5G and LPWAN for IoT solutions have been advocated. For instance, the authors in [10,19] opined that network congestion due to massive MTC and huge traffics will lower the QoS and IoT performance. They also advocated the need for a lightweight context-aware congestion control (CACC) mechanism that will enable IoT networks to mitigate the consequences of traffic congestion for better QoS.

The authors in [19] advocated that non-orthogonal multiple access (NOMA) for 5G access networks can be applied to support various quality of service (QoS) requirements for different types of end devices. They opined those technologies operating on licensed bands achieve QoS management, but the greatest disadvantage is cost due to licensed spectrum acquisition. The authors in [12,94,95] explained that NB-IoT employs a licensed spectrum and an LTE-based synchronous protocol, which are best for QoS at the expense of cost. Considering the tradeoff in QoS and cost, they revealed that NB-IoT is preferred for applications that require guaranteed QoS, but applications that do not have this constraint should choose LoRa or Sigfox. Sukhmani et al. [96] explained that network QoS and user QoE needs to be jointly considered. They illustrated the concept using a highly localized scenario, in which precise tracking of users and objects is critical. Thus, failing to locate objects precisely results in degraded QoE even though QoS requirements might be satisfied. Chen et al. [97] also pointed out that QoE may be further improved by emotional feedback according to where 5G and mobile cloud computing has reconsidered resource cognition and emotion-aware action feedback.

### 5.2. Cell Edge Interference Challenges and Solutions in 5G and LPWAN-IoT

There is cell edge interference in 5G networks and its impact on the end-user devices as the demand for public resources such as IoT, real-time video/virtual reality (VR) applications, and so on increases. Cell edge interference is the overlapping of signals with signals from other cell edge towers. This is bound to occur due to massive MTC in the 5G network. However, the problem of cell edge interference can be mitigated by implementing 5G jointly with a very long-range LPWAN such as LoRaWAN, Sigfox, Weightless-N, etc. Weightless-N is based on a television (TV) white space (WS) or TVWS standard, which operates in a Sub-GHz TV frequency band. It has up to 10 Mbps throughput with a range of over 20 km. Unused TV channels by the Primary Users (PU) are called White Space, which can be exploited by Cognitive Radio (CR) or TVWS technology to serve other devices or Secondary Users (SU). TVWS has a long propagation range with high penetration power rate properties and a high appreciable data rate. Hence, lightweight/low complexity TVWS like Weightless-N can be implemented with a novel strategy in the backhaul/last mile connectivity of a 5G network to overcome interference at the edge and IoT endpoint user. This will create room for wider dispersed cell edge towers deployment, thus, eliminating the closeness of towers that may lead to interference. Detailed research on TVWS is found in [98,99].

Furthermore, the problem of cell edge interference and its impact on the end-user in 5G has been investigated in literature from different perspectives. This problem occurs where adjacent cell towers direct radio frequencies (RF) across their target areas [20]. Some of the works involving cell edge interference include Abu-Mahfouz et al. [100], who opined that overlaying of small cells on macro-cells causes interference which affects small-cell edge users and NB-IoT UEs towards obtaining satisfactory QoS in the NB-IoT network. Muteba et al. [101] provided measures based on spectral efficiency, coverage, and capacity over heterogeneous infrastructures such as macro cells and small cells to reduce cell edge interference in NB-IoT.

Some mechanisms have also been proposed to address this problem. For instance, Wooseok et al. in [102] proposed advanced interference management for a 5G cellular network based on an elaborated joint scheduling mechanism. In [103], the authors developed a new scheduling technique to increase the probability of assigning the available resource blocks (RBs) to the cell-edge users so that their achieved throughput would increase. NOKIA in [104] developed a smart scheduler to mitigate cell-edge interference by employing the method of Frequency Selective Scheduling (FSS). FSS uses Channel Aware Scheduling (CAS) and Interference Aware Scheduling (IAS) to select non-faded RBs for each user. Lu Yang et al. [105] proposed a practical interference coordination scheme for 5G Cellular Networks based on beamforming and user scheduling. Also, citizen broadband radio service (CBRS) for 5G-IoT uses a strategy called spectrum access services (SAS) to mitigate interference from C-band (3.5–4.2 GHz) [106], though CBRS use cases are predominant in the United States (US).

## 6. Future Research Directions and Discussion

To enable billions of smart devices to interconnect autonomously and support massive MTC services and IoT applications, ubiquitous IoT connectivity is required to serve both urban and remote areas. This ubiquitous IoT connectivity is achieved by integrating the 5G network with LPWAN as discussed earlier. However, this 5G-LPWAN integration for ubiquitous use of IoT applications has introduced new challenges that need to be addressed. For instance, the 5G network is relatively deployed in mid-bands and in millimeter-wave bands, but these bands have medium and low ranges with low penetration rates. Hence, research involving a relative use of the 5G low band (Sub-1 GHz) is needed because the low band has a high penetration rate, long-range, and low data rate. These features are suitable for most IoT applications in cities and remote areas. The research in this sense should involve 5G architecture with mixed bands (low band, mid-band, and millimeter-wave band) according to the use case locations. For example, deployment of 5G low band in remote areas (rural communities) and some urban areas for most smart cities’ applications. Deployment of the 5G mid-band in the cities to support some IoT critical applications. Deployment of millimeter-wave in some cities area to support specifically huge bandwidth, ultra-reliable low latency (URLL), and mission-critical applications. The 5G network can serve as a backhaul connectivity solution to the IoT gateways. Another technology with a high penetration rate, long-range, and low data rate that operates at the Sub-1 GHz band is the IEEE 802.11 ah also called the Halow Wi-Fi. It is necessary to exploit the tremendous benefits of the Halow Wi-Fi device because of its varying technical requirements. These technical features include multiple channel bandwidth (CB) such as 1 Mhz, 2 Mhz, 4 Mhz, 8 Mhz, and 16 Mhz; and modulation and coding schemes (MCS) [6]. Hence, researchers can leverage these numerous features to provide varying IoT applications solution. For example, the use of 1 Mhz channel bandwidth will give a longer range, which can be used to support applications that require wider coverage. Likewise, the use of high MCS up to 256 QAM (quadrature amplitude modulation) will yield a very high throughput, which can be used to support high data rate IoT applications. Therefore, research in the Halow Wi-Fi technology is encouraged for improved smart city applications.

Overall, the findings and recommendations made with respect to backhaul connectivity solutions in 5G and LPWAN-IoT integration are:The use of 5G low band as backhaul connectivity between LPWAN-IoT gateway for low data rate IoT applications and remote areas’ applications;The use of 5G mid-band, Weightless-N (TVWS band), and Halow Wi-Fi device band as backhaul connectivity between LPWAN-IoT gateway for medium to high data rate IoT applications in a smart city scenario;The use of 5G millimeter-wave as backhaul connectivity between LPWAN-IoT gateway in cities area to support very high bandwidth, ultra-reliable low latency (URLL), and mission-critical IoT applications.

In addition, research is needed in mixed LPWAN devices to be integrated with 5G architecture for IoT applications. This will involve the deployment of appropriate LPWAN devices based on application types in certain areas. For example, LoRaWAN, Sigfox, NB-IoT, etc., for low-rate applications like smart meters; and Cat M1 and LTE-M for high bandwidth applications. Further, 5G-based IoT devices are recommended for smart cities’ applications. This will enable IoT applications to leverage the enormous benefits of the 5G network such as network slicing, AI/ML supported applications, mobile edge computing (MEC), enhanced security, and support for massive MTC including D2D communication. Hence, research in 5G IoT devices such as 5G eMTC, 5G NB2-IoT, and NR RedCap devices, including citizen broadband radio service (CBRS) as well as its devices (CBSD) is encouraged. This will help to boost the use cases of 5G eMTC, 5G NB-IoT, and 5G RedCap IoT in utilizing the tremendous benefit of the 5G network. Hence, the 5G RedCap IoT end device (ED) can be used to support medium to High bandwidth IoT applications that cannot be supported with conventional IoT EDs (LoRaWAN, Sigfox, NB-IoT). The CBRS is mainly in the United States (US) which is currently with many use cases. Other countries could harness comparable services such as a private 5G network with an equivalent 5G mid-band spectrum to that of the CBRS spectrum.

Also, a unified 5G and LPWAN-IoT architecture that will holistically support integration with emerging technologies and endogenous security is a possible new area of research. The authors believe that this unified architecture will go a long way to improve QoS/QoE and the security of smart cities and remote area applications in the 5G-LPWAN-IoT ecosystems. In addition, the joint consideration of QoS/QoE in an LPWAN-IoT application is a possible area of research.

Moreover, as earlier seen in Table 5, some countermeasures are highlighted to address some security challenges. These countermeasures are not enough to fully secure LPWAN-IoT due to the increased rate of new attacks and security breaches on IoT applications. Hence, hardening the security standards or requirements such as Confidentiality, Integration, Authenticity, and Availability (CIAA) will eliminate security breaches and vulnerabilities. This will make it difficult for an attacker to launch an attack on LPWAN-IoT solutions. Therefore, CIAA hardening is a possible area of research to safeguard IoT solutions. For instance, new approaches such as Machine Learning (ML) using supervised, unsupervised, or reinforcement learning algorithms, including Artificial Intelligence (AI) approach such as Artificial Neural Network (ANN) can be used for improved security standard solutions. This type of security solution has the potential to detect and prevent security occurrences based on attacker behavioral activities. Also, SDN and NFV are emerging areas of security standard hardening. This approach will improve the LPWAN-IoT ecosystem because most of the security mechanisms will be leveraged on the software level, thereby saving the device energy and complexity. Research in these areas for 5G and LPWAN solutions is encouraged.

Another possible area of research is in asymmetric cryptographic encryption algorithms for LPWAN-IoT. This is because most LPWANs are not supported by asymmetric encryptions due to their low computational complexity and low energy. Hence, lightweight asymmetric encryption is recommended for most non-cellular LPWAN security solutions This will provide stronger protection to massive MTC.

In addition, massive MTC will enable ubiquitous IoT. Over 80% of the earth is not covered by the internet. Thus, greater proportions of the earth should be covered by the internet for ubiquitous IoT to be actualized. For example, the ocean and forest areas should be covered for ocean vessel and wildlife tracking respectively. One of the new technologies that will support ubiquitous IoT connectivity is the low earth orbit (LEO) satellite, which is usually from 300 km to 2500 km above the earth’s surface. Researching on appropriate LPWAN jointly with LEO satellite for ubiquitous IoT connectivity is recommended as a very good area of research development. This will help for IoT availability even in the sea/ocean and remote rural areas. The LEO satellite constellation is a non-terrestrial network (NTN) solution, research in NTN is essential. Because compatibility issue abounds between the coexistence of terrestrial networks (TNs) with the non-terrestrial networks (NTNs) devices due to their dissimilarity physical (PHY) and media access (MAC) protocol layer properties. Hence, it is worthwhile to establish an appropriate frequency spectrum band for the TNs and NTNs devices. This will help to mitigate interference and the doppler effect caused by decibel (dB) isotropic losses.

Therefore, the use of 5G NB-IoT NTN is recommended for remote areas and sub-urban IoT applications. This is an emerging IoT solution that will help to address the problem of poor availability of cellular networks in these locations. Hence the 5G NB-IoT NTN will enable ubiquitous IoT connectivity, especially in rural developing countries that lack numerous IoT applications due to poor availability or out-of-reach of cellular networks.

Above all, a reliable 5G and LPWAN-IoT architecture for improved smart cities should support the following:Diversified backhaul connectivity;QoS/QoE support;Adequate integration with emerging technologies;Adequate interference management strategies;Endogenous security defense mechanism support;Ubiquitous or global connectivity support.

Hence, future research should be directed toward the aforementioned considerations for improved smart cities.

## 7. Conclusions

In this paper, various architectures involving 5G and LPWAN-IoT and Smart Cities are explored. This includes security challenges as well as endogenous security and solutions in 5G and LPWAN-IoT. The problem of Quality of Service (QoS)/Quality of Experience (QoE) in different application requirements and varying end-users are considered. The slicing of virtual networks based on different QoS to satisfy different services and QoE is discussed. Also, the problem of cell edge interference together with its impact on users in the 5G and LPWAN-IoT network is uncovered. A strategy involving the implementation of 5G jointly with Weightless-N (TVWS) technologies to reduce the cell edge interference is presented. Discussions on the need for ubiquitous connectivity using non-terrestrial networks (NTNs) such as LEO satellite integration with LPWAN-IoT, which resulted in a ubiquitous IoT solution called 5G NB-IoT NTN are presented. Also, a 5G IoT comprising 5G eMTC, 5G NB2-IoT, and 5G RedCap IoT device solution has been advocated in supporting varying IoT applications in the smart city ecosystem. Further, this paper also presents smart cities application grouping concerning various application technical requirements. this will enable an IoT designer to identify at a glance an appropriate LPWAN device that would be suitable for given IoT applications. In addition, future research directions which include recommendations such as the need for LEO satellite communications in ubiquitous IoT coverage; the use of ML, AI/ANN, SDN, and NFV for improved security solutions in IoT are discussed. This includes support for asymmetric cryptographic encryption algorithms for LPWAN-IoT. Furthermore, findings and recommendations are made with respect to some aspects such as backhaul connectivity for LPWAN-IoT, 5G RedCap IoT end device (ED) for medium to high bandwidth IoT applications, and 5G NB-IoT NTN for ubiquitous connectivity. Finally, a unified 5G and LPWAN-IoT architecture is discussed. The unified 5G architecture is expected to holistically support integration with emerging technologies and endogenous security for improved/secured smart cities and remote area IoT applications.

## Figures and Tables

**Figure 1 sensors-22-06313-f001:**
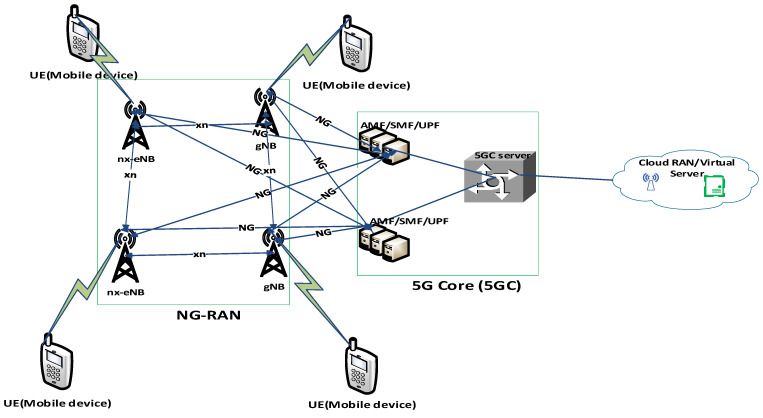
5G Network Architecture.

**Figure 2 sensors-22-06313-f002:**
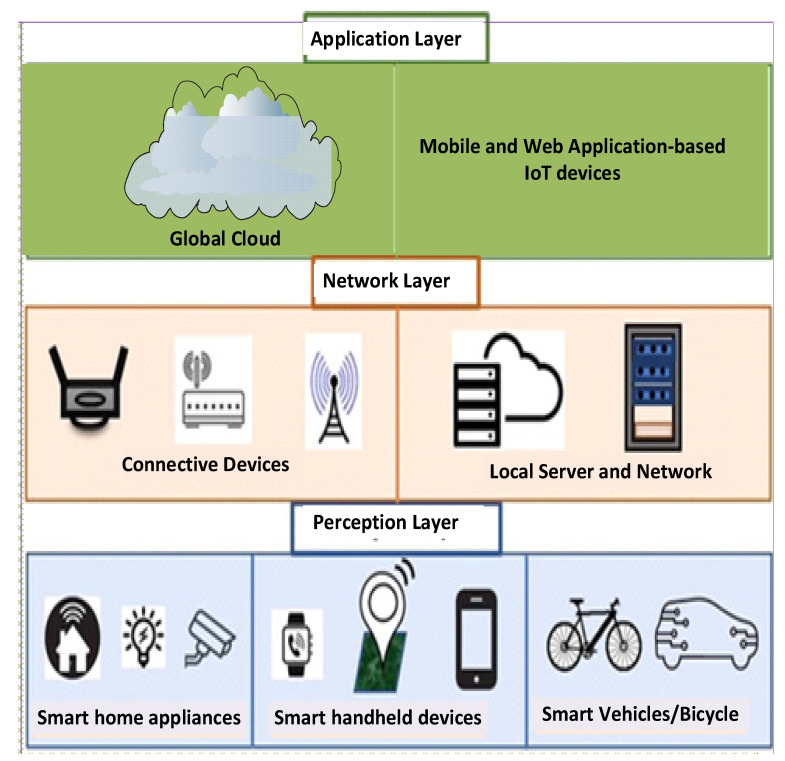
Three-Layer IoT Architecture (adapted from [38]).

**Figure 3 sensors-22-06313-f003:**
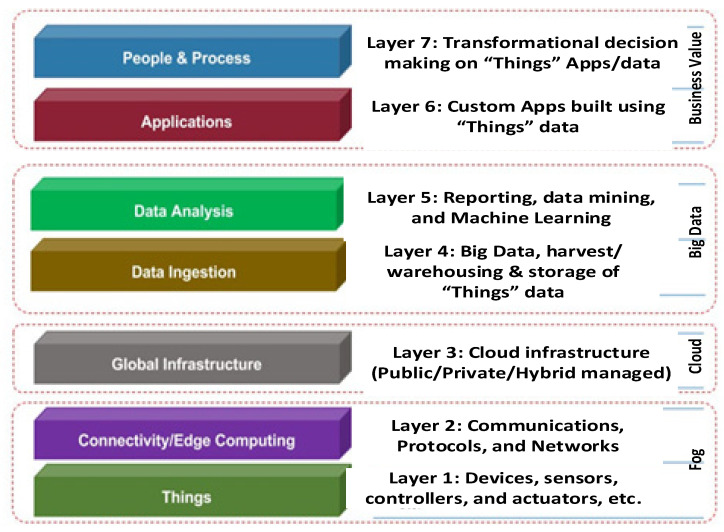
Seven-Layer IoT Architecture (adapted from [37]).

**Figure 4 sensors-22-06313-f004:**
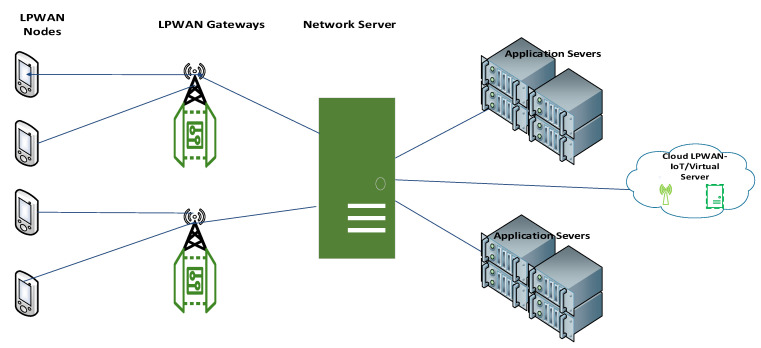
Typical LPWAN Architecture.

**Figure 5 sensors-22-06313-f005:**
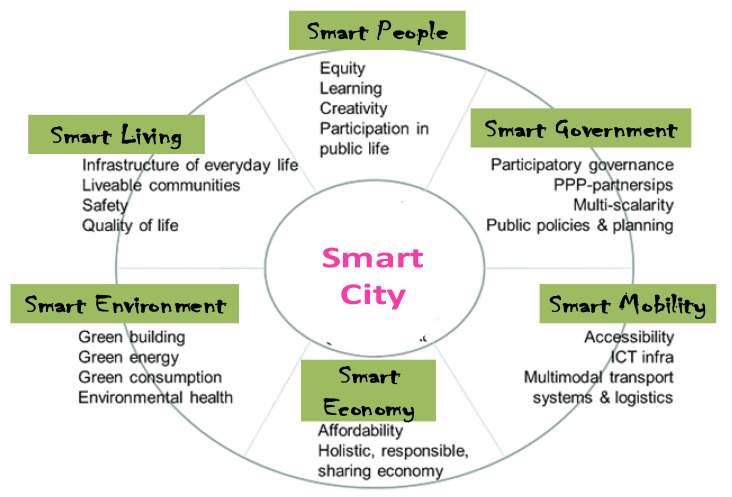
Smart City Wheel Conceptual Framework (adapted from [63,64]).

**Figure 6 sensors-22-06313-f006:**
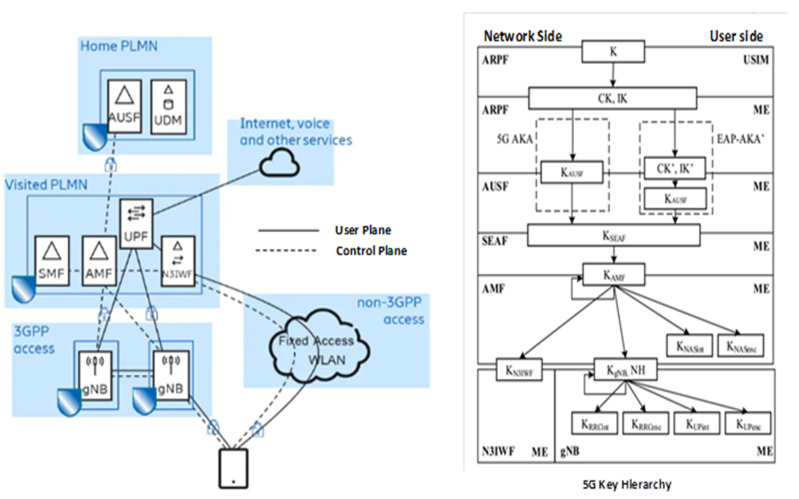
5G Basic Security Architecture and key hierarchy (adapted from [75]).

**Figure 7 sensors-22-06313-f007:**
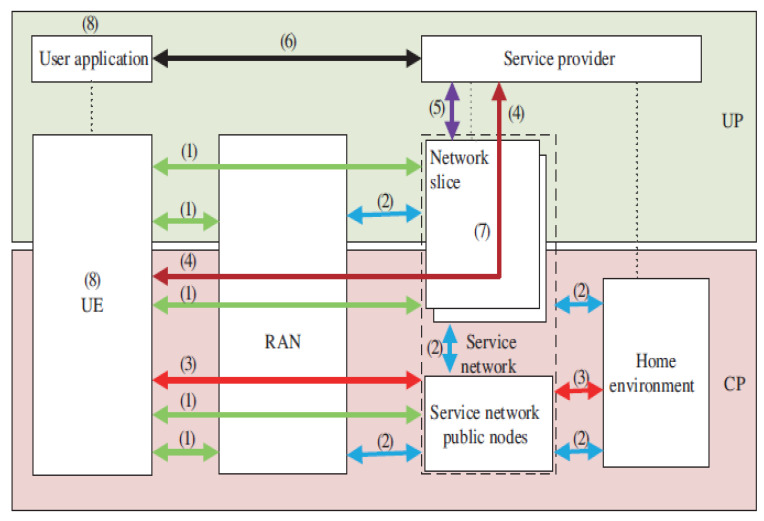
Proposed 5G Security by China IMT-2020 (5G) Promotion Group.

**Figure 8 sensors-22-06313-f008:**
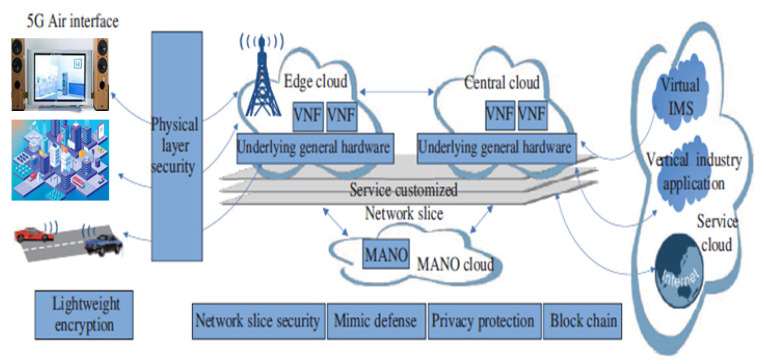
5G network endogenous defense security architecture (adapted from [26]).

**Table 1 sensors-22-06313-t001:** Comparisons of existing LPWAN-IoT Surveys with regards to consideration of the highlighted factors (Considered: √, Not considered: x).

Survey References	5G/LPWAN Integration	Endogenous Security	LEO Satellite	Cell-Edge Interference	QoS/QoE	Cryptographic Security	Smart-Cities Applications Grouping	5G-Based IoT (eMTC, NB2-IoT-Enhanced)
This Survey	5G: √, LPWAN: √	√	√	√	QoS: √, QoE: √	√	√	√
[1]	5G: √, LPWAN: √	x	x	x	QoS: √, QoE: x	x	x	X
[3]	5G: x, LPWAN: x	x	x	x	QoS: √, QoE: x	x	x	X
[4]	5G: x, LPWAN: x	x	x	x	QoS: x, QoE; x	x	x	X
[5]	5G: x, LPWAN: √	x	x	x	QoS: √, QoE; x	√	x	X
[8]	5G: √, LPWAN: √	x	x	x	QoS: √, QoE; x	√	x	X
[9]	5G: √, LPWAN: √	x	x	x	QoS: x, QoE; x	√	x	X
[10]	5G: √, LPWAN: √	x	x	x	QoS: √, QoE; x	√	x	X
[13]	5G: √, LPWAN: √	x	x	x	QoS: x, QoE; x	√	x	X
[14]	5G: √, LPWAN: √	x	x	x	QoS: √, QoE; x	√	x	X
[18]	5G: √, LPWAN: √	x	x	x	QoS: √, QoE; x	x	x	X
[19]	5G: √, LPWAN: √	x	x	x	QoS: √, QoE; x	√	x	X
[20]	5G: √, LPWAN: √	x	x	x	QoS: √, QoE; x	x	x	X
[22]	5G: √, LPWAN: √	x	x	x	QoS: √, QoE; x	x	x	X
[23]	5G: x, LPWAN: √	x	x	x	QoS: √, QoE; x	x	x	X
[24]	5G: √, LPWAN: √	x	x	x	QoS: √, QoE; x	√	x	X
[25]	5G: √, LPWAN: √	x	x	x	QoS: √, QoE; x	√	x	X
[27]	5G: √, LPWAN: √	x	x	x	QoS: √, QoE; x	x	x	X

**Table 2 sensors-22-06313-t002:** LPWAN Technologies Comparisons.

LPWAN Technologies Non-3GPP/3GPP	Frequency Spectrum	Latency	Throughput	Range (km)	Channel Bandwidth
LoRaWAN	<1 GHz	Low	≤40 kbps	30	≤500 kHz
Sigfox	<1 GHz	Low	≤150 bps	50	100 kHz
Dash	<1 GHz	Low	≤200 kbps	10	≤200 kHz
RPMA	2.4 GHz	Low	≤19,000 bps	20	80 MHz
Weightless-W	TVWS (≤900 MHz)	Low	≤10 Mbps	20	5 MHz
Weightless-N	<1 Ghz	Low	≤100 bps	5	200 Hz
Weightless-P	880–915 MHz	Low	≤100 kbps	4	≤100 kHz
LTE-M	455–2600 MHz	Very low	≤1 Mbps	5	1.44–5 MHz
5G eMTC	455–3500 MHz	Very low	≤2 Mbps	7	1.44–5 MHz
NB-IoT	455–2100 MHz	Very low	≤0.54 Mbps	7	180–200 kHz
5G NB2-IoT	455–3500 MHz	Very low	≤0.78 Mbps	10	≤500 kHz
LTE- Cat1	455–3500 MHz	Very low	≤10 Mbps	5	20 MHz
EC-GSM- IoT	395–1060 MHz	Very low	≤0.5 Mbps	8	≤500 kHz
IEEE 802.11ah	<1 Ghz	Very low	≤30 Mbps	2	≤16 MHz

**Table 3 sensors-22-06313-t003:** 3GPP Release numbers and Details.

3GPP Release	Release Date	Details
Phase 1	1992	Basic GSM
Phase 2	1995	GSM features including EFR Codec
Release 96	Q1 1997	GSM Updates, 14.4 kbps user data
Release 97	Q1 1998	GSM additional features, GPRS
Release 98	Q1 1999	GSM additional features, GPRS for PCS 1900, AMR, EDGE
Release 99	Q1 2000	3G UMTS incorporating WCDMA radio access
Release 4	Q2 2001	UMTS all-IP Core Network
Release 5	Q1 2002	IMS and HSDPA
Release 6	Q4 2004	HSUPA, MBMS, IMS enhancements, Push to Talk over Cellular, operation with WLAN
Release 7	Q4 2007	Improvements in QoS & latency, VoIP, HSPA+, NFC integration, EDGE Evolution
Release 8	Q4 2008	Introduction of LTE, SAE, OFDMA, MIMO, Dual Cell HSDPA
Release 9	Q4 2009	WiMAX / LTE / UMTS interoperability, Dual Cell HSDPA with MIMO, Dual Cell HSUPA, LTE HeNB
Release 10	Q1 2011	LTE-Advanced, Backwards compatibility with Release 8 (LTE), Multi-Cell HSDPA
Release 11	Q3 2012	Heterogeneous networks (HetNet), Coordinated Multipoint (CoMP), In device Coexistence (IDC), Advanced IP interconnection of Services,
Release 12	March 2015	Enhanced Small Cells operation, Carrier Aggregation (2 uplink carriers, 3 downlink carriers, FDD/TDD carrier aggregation), MIMO (3D channel modeling, elevation beamforming, massive MIMO), MTC—UE Cat 0 introduced D2D communication, eMBMS enhancements.
Release 13	Q1 2016	LTE-U/LTE-LAA, LTE-M, Elevation beamforming/Full Dimension MIMO, Indoor positioning, LTE-M Cat 1.4MHz & Cat 200kHz introduced
Release 14	Mid 2017	Elements on road to 5G
Release 15	End 2018	5G Phase 1 specification
Release 16	2020	5G Phase 2 specification
Release 17	~Sept 2021–June 2022	5G-based IoT ((eMTC, and NB2-IoT-enhanced) specifications completed. 5G RedCap completed.5G NB-IoT non-terrestrial networks (NTN) specifications.

**Table 4 sensors-22-06313-t004:** Smart Cities Application Groups.

Groups	Examples	Coverage	Bandwidth	Latency
(mMTC) smart cities applications	Smart utility meters (electricity, gas, and water meters), Smart homes, Smart buildings, Smart street light, Smart waste management, Smart car parking, Smart health care, E-Government & Smart public safety, Smart environment management, Smart retail, and supply chain	Medium/Long	Low	Low/High
(eMBB) smart cities applications	Virtual Reality (VR), Augmented Reality (AR), Tactile internet, Smart streaming, Smart Robotics, Smart Surveillance	Short	High/Very High	Very low
Critical infrastructure Smart Cities applications	Autonomous driving/connected cars, Industrial automation, Smart health monitoring, Smart traffic light control, Smart grid monitoring, Smart utility monitoring, Smart disaster monitoring, Smart asset monitoring, and fleet management, Smart structural monitoring, Smart oil and gas monitoring, Smart security and emergency/alarm, smart mobility	Medium/Long	Medium/High	Very low
Remote areas applications	Smart agriculture (agro-allied, farming, livestock, soil, and environmental measurement), Smart telemedicine, Smart distance education	Long	Low/High	Low/High

**Table 5 sensors-22-06313-t005:** Security threats and the applicable countermeasures in LPWAN-IoT.

Threats/Attacks	Concerned Security Requirements	Applicable Countermeasures	Applicable LPWAN
Replay attack	Confidentiality, Authenticity, Availability	Replay protection	LoRaWAN, Sigfox, and Weightless
Spoofing attack DoS attack	Authenticity, Confidentiality, and Integrity Availability and Authenticity	Authentication Network Monitor	Most LPWAN devices Most LPWAN devices
Wormhole	Availability and Authenticity	E2E security	Most LPWAN devices
Signal jamming Eavesdropping Man in the mid Floods attack	Availability and Integrity Confidentiality and Authenticity Confidentiality and Availability Authenticity, Availability, and Integrity	Secured credential Data Confidentiality Reliable delivery E2E security/monitor	LoRaWAN, Weightless, and NB-IoT LoRaWAN, Weightless, and DASH Most LPWAN devices Most LPWAN devices
Session hijacking Injections attack Sniffing attack Sybil attack	Availability and Authenticity Authenticity, Confidentiality, and Integrity Confidentiality Confidentiality, Authenticity, Availability	Certified equipment Updatability Network Monitor Updatability	LoRaWAN Most LPWAN devices Weightless and DASH Most LPWAN devices
Sinkhole attack	Availability and Authenticity	E2E security	Most LPWAN devices

**Table 6 sensors-22-06313-t006:** Cryptographic mechanisms and their encryption algorithms in 5G and LPWAN-IoT.

Cryptographic Mechanisms	Type of Algorithm	Concerned Security Requirements	Applicable LPWAN/5G
Symmetric Encryption	Advance encryption (AES)	Confidentiality	5G and Most LPWAN devices
Asymmetric Encryption	RSA/Elliptic Curve Crypto	Key management	5G and most 3GPP LPWANs
Key Agreement	Diffie-Hellman (DH)	Key agreement	5G, NB-IoT, LTE-M, Cat1, EC-GSM-IoT
Hashing Functions	SHA-1, SHA-2, and SHA-3	Integrity	5G, LTE-M, Cat1, NB-IoT, LoRaWAN
Digital Signature	Digital Signature Algorithm (DSA)	Digital signature	5G, LTE-M, and Cat1

## Data Availability

Not applicable.

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
