# Peer review of "A Survey on 5G and LPWAN-IoT for Improved Smart Cities and Remote Area Applications: From the Aspect of Architecture and Security"

_sensors, 2022, doi:10.3390/s22166313_

Round 1

Reviewer 1 Report

1. Introduction should be concise and precise. Highlight the contribution.

2. Previous studies are weak. Represent all previous studies in a table.

3. Lack of novel contribution.

4. Line 433: figure number is wrong.

5. Line 451: table 3 caption is not written in the correct place.

6. It looks like just an overview paper.

Author Response

Comments and Suggestions for Authors

  1. Introduction should be concise and precise. Highlight the contribution.

Authors’ Response: The authors thank and appreciate the reviewer’s comments.

We have made the introduction to be concise and clearer by highlighting key factors that are considered in the paper. The changes are made in red, as seen on page 3 (line 98 to 104).

The paper contributions have been re-written to reflect the focus of this survey. The changes are made in red as seen on page 3 (line 107 to 124). Our contributions are clearer now with the help of Table 1 which has just been added. This differentiates our work from existing surveys in IoT.  

  1. Previous studies are weak. Represent all previous studies in a table.

Authors’ Response: We have improved on the previous study by representing it in Table 1. Table 1 title is highlighted in red text as seen on page 5 (line 223-225).

  1. Lack of novel contribution.

Authors Response:  We have revised the title of the paper as a survey paper to depict the kind of expected contributions of the paper. The new table 1 is based on the highlighted key factors which show our novelty approach compared with the existing surveys that did not consider most of the highlighted factors. The details are found on page 5 (line 208 to 221). There are highlighted in red text.    

  1. Line 433: figure number is wrong.

Authors Response: We have corrected it. The changes are made in red.

  1. Line 451: table 3 caption is not written in the correct place.

Authors Response: This has also been corrected.

  1. It looks like just an overview paper.

Authors’ Response: The title of the paper has been revised to capture “Survey” in the title. We have also elaborated more on some aspects to give more technical information. The changes are made in red.

Reviewer 2 Report

The authors consider only LoRaWAN and other technologies as LPWAN, but the major number of this kind of technology has restriction by duty cycle and throughput. So, it is not clear why will be possible 5G give a huge benefit to LPWAN networks. 

It is necessary to present and consider IEEE 802.11ah.

The attacks and countermeasures on IoT need to be revised once it is not the focus of the paper. 

The title can be revised to reflect the main contribution of this paper.

The future directions for research need to be better explored. 

It is not clear the open challenges to conducting research on LPWAN-IoT over 5G.

It is necessary to improve the conclusion section by extending the text and giving more details about the benefits of this study.

To conclude, a better organization of the text needs to be conducted by the authors, increasing the possibility of this paper being cited and guiding new researchers in this field.

Author Response

Reviewer 2

Comments and Suggestions for Authors

The authors consider only LoRaWAN and other technologies as LPWAN, but the major number of this kind of technology has restriction by duty cycle and throughput. So, it is not clear why will be possible for 5G to give a huge benefit to LPWAN networks. 

Authors’ Response: The authors thank and appreciate the reviewer’s comments.

The authors have now considered the IEEE 802.11ah standard as one of the LPWAN technologies. It is highlighted in red in the main text as seen on page 12 (line 457 to 466). Also, because of the duty cycle of LPWAN, we have now revised the section of the title to be “5G and LPWAN-IoT” instead of the earlier “ 5G enabled LPWAN-IoT”.it is highlighted in red in the title.

We have also clarified how 5G will benefit LPWAN-IoT, by indicating that 5G can serve as backhaul connectivity to LPWAN gateways. This is highlighted in red on page 2 (line 76 to 77).  Also, we have shown that 5G can be leveraged to support massive MTC, which is one of the key features of the 5G network standard. The details with references are found on page 2 (line 56 to 61) in red.

It is necessary to present and consider IEEE 802.11ah.

Authors’ Response: We have now presented and considered the IEEE 802.11ah standard including a discussion on this in the “Future research directions” section. The changes are made in red on page 12 (line 457 to 466) and page 25 (line 970 to 982).

The attacks and countermeasures on IoT need to be revised once it is not the focus of the paper. 

Authors’ Response: It is part of the focus of the paper since the paper considers security challenges and solutions in the title.

The title can be revised to reflect the main contribution of this paper.

Authors’ Response: the title has now been revised as “A survey on 5G and LPWAN-IoT for Improved Smart Cities and Remote Area Applications: Architecture, Security, Challenges, and Solutions” to reflect the main contribution of the paper.

The future directions for research need to be better explored. 

Authors’ Response: we have now further explained the future directions for research with the addition of numerous aspects. The changes are highlighted in red on pages 25 to 26.

It is not clear the open challenges to conducting research on LPWAN-IoT over 5G.

Authors’ Response: We have clarified this, by changing the title to now bear 5G and LPWAN-IoT. However, our emphasis includes 5G serving as an enabling technology for LPWAN-IoT. For example, we have included 5G IoT such as 5G eMTC, 5G NB-IoT, and 5G RedCap IoT devices. The changes are made in red on page 3 (line 98 to 104).

It is necessary to improve the conclusion section by extending the text and giving more details about the benefits of this study.

Authors’ Response: We have improved on the conclusion with more details that are beneficial. The changes are made in red on page 27 (line 1049 to 1057)

To conclude, a better organization of the text needs to be conducted by the authors, increasing the possibility of this paper being cited and guiding new researchers in this field.

Authors’ Response: We have well-organized the texts in this paper.

Round 2

Reviewer 1 Report

1. This paper just overviewed some existing manuscrpts, what is the new contribution?

2. What are the findings?

Author Response

Comments and Suggestions for Authors

  1. This paper just overviewed some existing manuscripts, what is the new contribution?

Authors Response: The authors thank and appreciate the reviewer’s comments.

In order to clarify our contributions, we have provided the sections where the details that support each contribution are found:

Contribution (1) in (line 107 to 109): Categorize smart cities’ applications based on the finding from the exploration and investigation of 5G and LPWAN-IoT with respect to emerging technologies.

Supported details: Section 3.4.1, in page 14-15 (line 493 to 512).

Contribution (2) in (line 110 to 113): Identify research gaps in the aspect of security based on intensive investigation of security challenges in 5G and LPWAN-IoT. Research gaps such as the need of applying endogenous and cryptographic security in 5G and LPWAN-IoT are unarguable.

Supported details: On pages 21 to 22 (line 788 to 842).

Contribution (3) in (line 114 to 116): Advocate the remediation of cell-edge interference problem using a special television white space (TVWS) strategy as backhaul connectivity for LPWAN-IoT solutions in a smart city scenario.

Supported details: On page 24 (line 889 to 905).

Contribution (4) In (line 117 to 120): Determine adequate strategy based on emerging technology for joint consideration of the quality of service (QoS)/quality of experience (QoE) in different application requirements and varying end users in 5G and LPWAN-IoT. In this respect, QoS and QoE are considered jointly in an IoT application.

Supported details: Emerging technologies such as ML/AL, Network slicing, and SDN/NFV are identified for possible joint consideration of QoS/QoE in IoT applications; as seen on page 23 (line 849 to 869) and on page 26 (line998 to 1003).

Contribution (5) In (line 121 to 124): Identify optimal technologies that will address the ubiquity connectivity bottleneck in the LPWAN-IoT ecosystem. For example, the use of non-terrestrial networks (NTNs) such as low-earth orbit (LEO) satellite constellations integration in LPWAN-IoT is advocated.

 Supported details: Section 5.3, on pages 24-25 (line 930 to 952).

  1. What are the findings?

Authors Response: Findings and recommendations in some aspects are now added as found on:

  • pages 25-26 in (line 983 to 992): “Overall, the findings and recommendations made with respect to backhaul connectivity solutions in 5G and LPWAN-IoT integration are
  • The use of 5G low band as backhaul connectivity between LPWAN-IoT gateway for low data rate IoT applications and remote areas’ applications.
  • The use of 5G mid-band, Weightless-N (TVWS band), and Halow Wi-Fi de-vice band as backhaul connectivity between LPWAN-IoT gateway for medium to high data rate IoT applications in a smart city scenario.
  • The use of 5G millimeter-wave as backhaul connectivity between LPWAN-IoT gateway in cities area to support very high bandwidth, ultra-reliable low latency (URLL), and mission-critical IoT applications.”
  • Page 26 in (line 1004-1007): “Thus, the 5G RedCap IoT end device (ED) can be used to support medium to High bandwidth IoT applications that cannot be supported with conventional IoT EDs (LoRaWAN, Sigfox, NB-IoT).”
  • Page 27 in (line 1052-1056): “Therefore, the use of 5G NB-IoT NTN is recommended for remote areas and sub-urban IoT applications. This is an emerging IoT solution that will help to address the problem of poor availability of cellular networks in these locations. Hence the 5G NB-IoT NTN will enable ubiquitous IoT connectivity, especially in rural developing countries that lack numerous IoT applications due to poor availability or out-of-reach of cellular networks.”
  • Page 27 in (line 1077-1083), Conclusion Section: “Furthermore, findings and recommendations are made with respect to some aspects such as backhaul connectivity for LPWAN-IoT, 5G RedCap IoT end device (ED) for medium to high bandwidth IoT applications, and 5G NB-IoT NTN for ubiquitous connectivity.”

Submission Date

02 June 2022

Date of this review

27 Jul 2022 10:31:03

Reviewer 2 Report

Now, the paper can be accepted in its present form.

Author Response

Comments and Suggestions for Authors

Now, the paper can be accepted in its present form.

Authors Response: Thank you so much for the acceptance.

Submission Date

02 June 2022

Date of this review

27 Jul 2022 20:26:39

Round 3

Reviewer 1 Report

The authors have revised this manuscript according to my previous review report. Now, I agree to accept this manuscript in present form.